# Learning to Decode: Reinforcement Learning for Decoding of Sparse Graph-Based Channel Codes

**Salman Habib†, Allison Beemer\*, and Jörg Kliewer†**
†Helen and John C. Hartmann Dept. of Electrical and Computer Engineering,
New Jersey Institute of Technology, Newark, NJ 07102
\*Dept. of Mathematics, University of Wisconsin-Eau Claire, Eau Claire, WI 54701
sh383@njit.edu,beemera@uwec.edu,jkliewer@njit.edu

## Abstract

We show in this work that reinforcement learning can be successfully applied to decoding short to moderate length sparse graph-based channel codes. Specifically, we focus on low-density parity check (LDPC) codes, which for example have been standardized in the context of 5G cellular communication systems due to their excellent error correcting performance. These codes are typically decoded via belief propagation iterative decoding on the corresponding bipartite (Tanner) graph of the code via flooding, i.e., all check and variable nodes in the Tanner graph are updated at once. In contrast, in this paper we utilize a sequential update policy which selects the optimum check node (CN) scheduling in order to improve decoding performance. In particular, we model the CN update process as a multi-armed bandit process with dependent arms and employ a Q-learning scheme for optimizing the CN scheduling policy. In order to reduce the learning complexity, we propose a novel graph-induced CN clustering approach to partition the state space in such a way that dependencies between clusters are minimized. Our results show that compared to other decoding approaches from the literature, the proposed reinforcement learning scheme not only significantly improves the decoding performance, but also reduces the decoding complexity dramatically once the scheduling policy is learned.

## 1   Introduction

Binary low-density parity-check (LDPC) codes [10] are sparse graph-based channel codes that have recently been standardized for data communication in the 5G cellular new radio standard due to their excellent error correcting performance [5, 15]. In practice, LDPC codes are iteratively decoded via *belief propagation* (BP), an algorithm that operates on the Tanner graph of the code [26]. Tanner graphs of LDPC codes are sparse bipartite graphs whose vertex sets are partitioned into two types: check nodes (CNs) and variable nodes (VNs). Typically, iterative decoding on a Tanner graph is carried out via flooding, i.e., all CNs and VNs are updated simultaneously [9]. However, in [4], a sequential CN scheduling scheme, so-called node-wise scheduling (NS), was proposed. In this scheme, messages are sent from a single CN at a time, and successive CNs are scheduled based on their residuals, i.e., the magnitude of the difference between subsequent messages emanating from the CN.

NS of an iterative decoder can lead to improved performance, as was shown in [4]. The implementation relies on the intuition that in loopy BP, the higher the residual of a CN, the further away that portion of the graph is from convergence. Hence, scheduling CNs with higher residuals is expected to lead to faster decoder convergence. Although the NS method converges faster than the flooding scheme, the computation of provisional messages for updating the CN residuals in real-time is

required, rendering it more computationally intensive than flooding for the same total number of messages propagated.

To mitigate the computational complexity inherent in NS, we propose a multi-armed bandit-based NS (MAB-NS) scheme for sequential iterative decoding of short LDPC codes.[1] Rather than computing residuals, MAB-NS evaluates an action-value function prior to scheduling, which determines how beneficial an action is for maximizing convergence speed and performance of the decoder. An action is defined here as selecting a single CN to convey its outgoing messages to its adjacent variable nodes. The NS algorithm is modeled as a Markov decision process (MDP) [25], where the Tanner graph is viewed as an $m$-armed slot machine with $m$ CNs (arms), and an agent learns to schedule CNs that elicit the highest reward. Repeated scheduling in the learning phase enables the agent to estimate the action-value function, where the optimal scheduling order is the one that yields a codeword output by propagating the smallest number of CN to VN messages.

Neural network assisted decoding of linear codes codes has been addressed in, e.g., [19, 14, 13, 1], aiming to learn the noise on the communication channel. However, these methods have not been applied to sparse graph-based codes and, due to their high learning complexity, are suited only for codes with short block lengths. Reinforcement learning (RL) has been applied recently for hard decision based iterative decoding in [3]. Further, a deep learning framework based on hyper-networks was used for BP decoding of short block length LDPC codes in [20], where the hidden layers of the network unfold to represent factor graphs executing successive message passing iterations. However, to the best of our knowledge, reinforcement learning (RL) has not been successfully applied to iterative decoding of LDPC codes in the open literature so far.

The main ingredient for the proposed RL decoder is Q-learning [28, 29]. Q-learning is a Monte Carlo approach for estimating the action-value function without explicit assumptions on the distribution of the bandit processes [7]. However, a major drawback of applying Q-learning to the sequential CN scheduling problem at hand is that the learning complexity grows exponentially with the number of CNs. A straightforward way to select the underlying state space of the Q-learning problem would be to consider the vector of quantized CN values. However, for practical LDPC codes, the number of CNs ranges in the hundreds, and even for a binary quantization of each CN value the cardinality of the state space is not computationally manageable in the learning process. A multitude of methods for reducing the learning complexity in RL have been proposed in the literature: for example, complexity may be reduced by partitioning the state space (see, e.g., [24, 18]), imposing a state hierarchy (see, e.g., [21]), or by dimensionality reduction (see, e.g., [2, 23]).

In this work, we follow a similar avenue, albeit tailored to the problem at hand. In extension of our previous work [12], for Q-learning we propose grouping the CNs into clusters, each with a separate state and action space. While this approach has the potential to make learning tractable, it also assumes independence of the clusters; this assumption cannot hold due to the presence of cycles in the Tanner graph. In order to mitigate the detrimental effect of clustering on the learning performance, we propose to leverage the structure of the Tanner graph of the code. In particular, we aim to optimize the clusters in such a way that dependencies between them are minimized. Note that as NS follows a fixed greedy schedule, there exists a non-zero probability that initially correct, but unreliable, bits are wrongly corrected into an error that is propagated in subsequent iterations. In contrast, our proposed scheme based on Q-learning allows some room for exploration by scheduling the CN with the highest expected *long-term* residual, mitigating such a potential error propagation.

To this end, we define novel graphical substructures in the Tanner graph termed cluster connecting sets, which capture the connectivity between CN clusters, and analyze their graph-theoretic properties. Numerical results show that RL in combination with sequential scheduling provides a significant improvement in the performance of LDPC decoders for short codes, superior to existing approaches in the literature.

# 2 Preliminaries

## 2.1 Low-density parity-check codes

An $[n, k]$ binary linear code is a $k$-dimensional subspace of $\mathbb{F}_2^n$, and may be defined as the kernel of a (non-unique) binary *parity-check matrix* $\mathbf{H} \in \mathbb{F}_2^{m \times n}$, where $m \geq n - k$. The *Tanner graph* of a linear code with parity-check matrix $\mathbf{H}$ is the bipartite graph $G_{\mathbf{H}} = (V \cup C, E)$, where $V = \{v_0, \ldots, v_{n-1}\}$ is a set of variable nodes (VNs) corresponding to the columns of $\mathbf{H}$, $C = \{c_0, \ldots, c_{m-1}\}$ is a set of check nodes (CNs) corresponding to the rows of $H$, and edges in $E$ correspond to the 1's in $\mathbf{H}$ [26] (see an example in Fig. 1 below). That is, $\mathbf{H}$ is the (simplified) adjacency matrix of $G_{\mathbf{H}}$. LDPC codes are a class of highly competitive linear codes defined via sparse parity-check matrices or, equivalently, sparse Tanner graphs [10]. Due to this sparsity, LDPC codes are amenable to low-complexity graph-based message-passing decoding algorithms, making them ideal for practical applications. BP iterative decoding, considered here, is one such algorithm.

In this work, we present experimental results for two particular classes of LDPC codes: $(j, k)$-regular and array-based (AB-) LDPC codes. A $(j, k)$-regular LDPC code is defined by a parity-check matrix with constant column and row weights equal to $j$ and $k$, respectively [10]. A $(\gamma, p)$ AB-LDPC code, where $p$ is prime, is a $(\gamma, p)$-regular LDPC code with additional structure in its parity-check matrix, $\mathbf{H}(\gamma, p)$ [8]. In particular,

$$\mathbf{H}(\gamma, p) = \begin{bmatrix} \mathbf{I} & \mathbf{I} & \mathbf{I} & \cdots & \mathbf{I} \\ \mathbf{I} & \sigma & \sigma^2 & \cdots & \sigma^{p-1} \\ \vdots & \vdots & \vdots & \cdots & \vdots \\ \mathbf{I} & \sigma^{\gamma-1} & \sigma^{2(\gamma-1)} & \cdots & \sigma^{(\gamma-1)(p-1)} \end{bmatrix}, \tag{1}$$

where $\sigma^z$ denotes the circulant matrix obtained by cyclically left-shifting the entries of the $p \times p$ identity matrix $\mathbf{I}$ by $z \pmod{p}$ positions. Notice that $\sigma^0 = \mathbf{I}$. Each row (resp. column) of submatrices of $\mathbf{H}(\gamma, p)$ forms a *row* (resp. *column*) *group*. Observe that there are a total of $p$ (resp. $p^2$) column groups (resp. columns) and $\gamma$ (resp. $\gamma p$) row groups (resp. rows) in $\mathbf{H}(\gamma, p)$.

## 2.2 Multi-armed bandits

The MAB problem is a special RL problem where in each time step, a gambler must decide which arm of an $m$-armed slot machine to pull in order to maximize the total reward in a series of pulls. We model the $m$-armed slot machine as a MDP. In an MDP, a learner must decide which action to take at each time step by observing only the current state of its environment. This decision-making process leads to a future state and expected reward that are contingent only on the current state and action. For finite state and action spaces, the process is called a finite MDP [25], and the optimized scheduling of arms (CNs in our case) is obtained by solving the MAB problem.

In the remainder of the paper, let $[[x]] \triangleq \{0, \ldots, x - 1\}$, where $x$ is a positive integer. In an $m$-armed bandit problem, let $S_t^{(0)}, \ldots, S_t^{(m-1)}$ represent $m$ bandit processes (arms) at time $t$, where each random variable (r.v.) $S_t^{(j)}$, $j \in [[m]]$, can take $M$ possible real values. Let a state space $\mathcal{S}^{(M)}$ contain all $M^m$ possible realizations of the sequence $S_t^{(0)}, \ldots, S_t^{(m-1)}$, and let the r.v. $S_t$, with realization $s \in [[M^m]]$ represent the index of realization $s_t^{(0)}, \ldots, s_t^{(m-1)}$. Since each index corresponds to a unique realization, we also refer to $S_t$ as the state of the $m$-armed slot machine at time $t$. If the arms are modeled as independent bandit processes, we define a r.v. $\hat{S}$, with realization $\hat{s} \in [[M^m]]$, as the realization $s_t^{(j)}$ of a particular arm $j$. Let $A_t$ represent an action, with realization $a \in \mathcal{A} := [[m]]$, indicating the index of an arm that has been pulled by the gambler at time $t$. Let $S_{t+1}$ represent the new state of the MDP after pulling arm $A_t$, and let $s'$ denote its realization. Also, let a r.v. $R_t(S_t, A_t, S_{t+1})$, with realization $r$ be the reward yielded at time $t$ after playing arm $A_t$, in state $S_t$ that yields state $S_{t+1}$.

## 2.3 MAB-NS scheme

In the following, we propose a MAB-NS scheme as a sequential BP decoding algorithm. Specifically, a single message passing iteration is given by messages sent from a single CN to all its neighboring VNs, and subsequent messages sent from these VNs to their remaining CN neighbors. Sequential CN scheduling is carried out until a stopping condition is reached, or an iteration threshold is exceeded. The MAB-NS decoder applies a scheduling policy, based on an action-value function, avoiding the computationally costly real-time calculation of residuals.

The NS algorithm informs an imaginary agent of the current state of the decoder and the reward obtained after performing an action (scheduling a CN). Based on these observations, the agent takes future actions, to enhance the total reward earned, which alters the state of the environment as well as the future reward. In MAB-NS, the reward $R_a$ obtained by the agent after scheduling CN $a$ is defined by $R_a = \max_{v \in \mathcal{N}_V(a)} r_{a \to v}$, where the residual $r_{a \to v}$ is computed according to

$$r_{a \to v} \triangleq |m'_{a \to v} - m_{a \to v}|. \tag{2}$$

Here, $m_{a \to v}$ is the message sent by CN $a$ to its neighboring VN $v$ in the previous iteration, and $m'_{a \to v}$ is the message that CN $a$ would send to VN $v$ in the current iteration, if scheduled.

In the proposed MAB problem, the decoder iteration $\ell$ is analogous to a time step $t$. Let $\mathbf{x} = [x_0, \ldots, x_{n-1}]$ and $\mathbf{y} = [y_0, \ldots, y_{n-1}]$ represent the transmitted and the received codeword, resp., where $x_i \in \{0, 1\}$, and $y_i = (-1)^{x_i} + z$ with $z \sim \mathcal{N}(0, \sigma^2)$. Let $\hat{\mathbf{x}} \in \mathbb{F}_2^n$ be the reconstructed codeword by the decoder. The posterior log-likelihood ratio (LLR) of $x_i$ is expressed as $L_i = \log \frac{\Pr(x_i=1|y_i)}{\Pr(x_i=0|y_i)}$. The soft channel information input to the MAB-NS algorithm is a vector $\mathbf{L} = [L_0, \ldots, L_{n-1}]$ comprised of LLRs. In the MAB-NS scheme, the state of CN $j$ at the end of iteration $\ell$ is given by $\hat{s}_\ell^{(j)} = \sum_{i=0}^{n-1} H_{j,i} \hat{L}_\ell^{(i)}$, where $\hat{L}_\ell^{(i)} = \sum_{c \in \mathcal{N}(v_i)} m_{c \to v_i} + L_i$ is the posterior LLR computed by VN $v_i$ at the end of iteration $\ell$, and $m_{c \to v_i}$ is the message received by VN $v_i$ from neighboring CN $c$. Let $\hat{\mathbf{S}}_\ell = \hat{S}_\ell^{(0)}, \ldots, \hat{S}_\ell^{(m-1)}$, with realization $\hat{\mathbf{s}}_\ell = \hat{s}_\ell^{(0)}, \ldots, \hat{s}_\ell^{(m-1)}$, represent a soft syndrome vector of the MAB-NS scheme obtained at the end of iteration $\ell$. This soft syndrome vector represents the state of the decoder in each iteration.

It is necessary to quantize each CN state in order to obtain a finite-cardinality state space. Let $g_M(\cdot)$ denote an $M$-level scalar quantization function that maps a real number to any of the closest $M$ possible representation points set by the quantizer. Then, $\mathbf{S}_\ell = S_\ell^{(0)}, \ldots, S_\ell^{(m-1)}$ is the quantized syndrome vector, where a realization is given as $s_\ell^{(j)} = g_M(\hat{s}_\ell^{(j)})$.

The proposed MAB-NS scheme is shown in Algorithm 1, omitted here for space reasons, but stated in the supplementary material Section S.2. The algorithm inputs are a soft channel information vector $\mathbf{L}$ and a parity-check matrix $\mathbf{H}$; the output is the estimated (decoded) codeword $\hat{\mathbf{x}}$. Note that the time complexity for selecting a CN in line 12 of Algorithm 1 grows linearly with the total number of CNs, as opposed to being zero for flooding BP.

We also note that the MAB-NS algorithm is dynamic and depends both on the graph structure and on received channel values: thus, the scheduling order adapts with subsequent transmissions, and will outperform a scheduling order which is fixed in advance.

## 2.4 Solving the MAB problem using clustered Q-learning

Q-learning is an adaptive algorithm for computing optimal policies for MDPs that does not rely on explicit assumptions on the distribution of the bandit processes. As discussed above, in a traditional Q-learning approach, the state space observed by the agent grows exponentially with the number of arms, $m$. In order to reduce both the state space and the learning complexity, we propose a novel clustering strategy in the following. A cluster is defined as a set of arms with its own state and action spaces. Let $z$ represent the size of (number of arms in) a cluster. The state of a cluster with index $u \in [[\lceil \frac{m}{z} \rceil]]$ at iteration $\ell$ is a sub-syndrome $\mathbf{S}_\ell^{(u,z)} = S_\ell^{(uz)}, \ldots, S_\ell^{(uz+z-1)}$ of the syndrome $\mathbf{S}_\ell = S_\ell^{(0)}, \ldots, S_\ell^{(m-1)}$, with a state space $\mathcal{S}_u^{(M)}$ containing all possible $M^z$ sub-syndromes $\mathbf{S}_\ell^{(u,z)}$. Hence,

the total number of states that may be observed by the agent is upper-bounded by $\left\lceil \frac{m}{z} \right\rceil |\mathcal{S}_u^{(M)}|$. Notice that as long as $z \ll m$, $|\mathcal{S}_u^{(M)}| \ll |\mathcal{S}^{(M)}|$. The action space of cluster $u$ is defined as $\mathcal{A}_u = [[z]]$. The action-value function for cluster $u$ in the $(\ell + 1)$st iteration is given by

$$Q_{\ell+1}(s_u, a_u) = (1-\alpha)Q_\ell(s_u, a_u) + \alpha\Big(R_\ell(s_u, a_u, f(s_u, a_u)) + \beta \max_{u', a_{u'}} Q_\ell(f(s_{u'}, a_{u'}), a_{u'})\Big), \quad (3)$$

where $s_u \in [[M^z]]$ and $a_u \in \mathcal{A}_u$ are the state and action indices, respectively, for cluster $u$, $f(s_u, a_u)$ represents the new cluster state $s'_u \in [[M^z]]$ as a function of $s_u$ and $a_u$, $R_\ell(s_u, a_u, s'_u)$ is the reward obtained after taking action $a_u$ in state $s_u$, and $0 \leq \alpha \leq 1$ is the learning rate. In clustered Q-learning, the action $a_u$ in time step $\ell$ is selected via an $\epsilon$-greedy approach according to

$$a_u = \begin{cases} u \text{ and } \mathcal{A}_u \text{ selected uniformly at random w.p. } \epsilon, \\ \pi_Q^{(\ell)} \text{ selected w.p. } 1 - \epsilon, \end{cases} \quad (4)$$

where $\pi_Q^{(\ell)} = \arg\max_{a_u \text{ s.t. } u \in [[\lceil \frac{m}{z} \rceil]]} Q_\ell(s_u, a_u)$. The action-value function is recursively updated $\ell_{\max}$ times according to (3), where $\ell_{\max}$ is the maximum number of times a CN is scheduled. The detailed proposed clustered Q-learning scheme is given in Algorithm 2 in the supplementary material Section S.3.

The input to Algorithm 2 is a set $\mathscr{L} = \{\mathbf{L}_0, \ldots, \mathbf{L}_{|\mathscr{L}|-1}\}$ containing $|\mathscr{L}|$ realizations of $\mathbf{L}$ over which clustered Q-learning is performed, and a parity-check matrix $\mathbf{H}$. This algorithm trains an agent to learn the optimum CN scheduling policy for a given $\mathbf{H}$. In each optimization step $\ell$, the agent performs NS for a given $\mathbf{L}$. As a result, clustered Q-learning can be viewed as a recursive Monte Carlo estimation approach with $\mathbf{L}$ as the source of randomness.

# 3  Cluster-connecting Sets

In clustered Q-learning, the total number of states observed by the agent is upper-bounded by $\left\lceil \frac{m}{z} \right\rceil |\mathcal{S}_u^{(M)}| \ll |\mathcal{S}^{(M)}|$. Note that the learning complexity in clustered Q-learning is $\mathcal{O}(z^M)$, whereas in standard Q-learning the complexity scales as $\mathcal{O}(m^M)$, with the cluster size $z \ll m$. Consequently, clustering enables a tractable RL task for sequential scheduling of short LDPC codes.

On the other hand, the larger the size of the cluster, the greater the ability of the agent to take into account any dependencies between CN log-likelihood ratios (LLRs). Hence, there exists a trade-off between the effectiveness of clustered Q-learning and the learning complexity. Indeed, cluster size should be large enough to accommodate the dependencies between CN messages as much as possible, without being so large that learning is infeasible. We note that clustered Q-learning serves as an approximation for standard Q-learning, but suffers from a performance loss due to the existence of dependencies between CN messages emanating from separate clusters.

To better understand this loss in performance, we introduce a critical sub-structure in the Tanner graph $G_{\mathbf{H}}$ of an LDPC code. Let $C_u = \{c_1, \ldots, c_z\}$ denote the set of CNs belonging to the cluster of index $u$, and let $\bar{C}_u = C \setminus C_u$ be the remaining CNs in the Tanner graph. Let $\mathcal{N}_V(C_u)$ be the set of all neighbors of $C_u$ in $V$. For $W \subseteq \mathcal{N}_V(C_u)$, let $\mathcal{N}_C(W)$ be the set of all neighbors of $W$ in $C$.

**Definition 1.** *Fix a CN cluster $C_u$, and let $W \subseteq \mathcal{N}_V(C_u)$. We say that $W$ is the cluster-connecting set (CCS) corresponding to $C_u$ if the following holds: $v \in W$ if and only if $\mathcal{N}_{C_u}(v)$ and $\mathcal{N}_{\bar{C}_u}(v)$ are both nonempty. If $|W| = A$, and $|\mathcal{N}_{C_u}(W)| = B$, we say that $W$ is an $(A, B)$ CCS.*

CCSs induce dependencies between the messages generated by the CNs in $C_u$ and those in $\bar{C}_u$. Roughly speaking, the number of edges in a Tanner graph that connect $C_u$ to $\bar{C}_u$ grows with the size of $C_u$'s CCS. That is, on average, the larger the size of the CCS corresponding to $C_u$, the greater the dependence between $C_u$ and $\bar{C}_u$. This holds absolutely for Tanner graphs in which all VNs have the same degree.

Fig. 1 depicts an example of a Tanner graph $G_{\mathbf{H}}$ containing a cluster of CNs whose corresponding CCS, $W$, is a $(3, 3)$ CCS. Note that in this example there are no cycles in the subgraph induced by $W \cup \mathcal{N}_{C_u}(W)$. However, there are multiple 4-cycles in the subgraph induced by $(\mathcal{N}_V(C_u) \setminus W) \cup C_u$.

In the remainder of the paper, we assume all Tanner graphs are connected and contain cycles. We first show that any choice of cluster that is a proper subset of the CNs has a nonempty CCS.

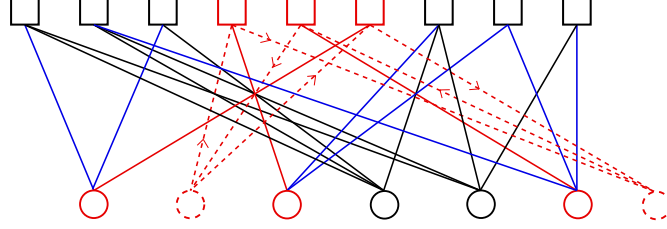

**Figure 1:** Depiction of the $(3, 3)$ CCS (shown using solid red circles) corresponding to cluster $C_u$ (red squares). The dashed edges connect $C_u$ to neighboring VNs (dashed red) which do not belong to the CCS. Arrows highlight the cycles incident to the cluster.

**Proposition 1.** *Consider a Tanner graph with $m$ CNs, and choose a cluster $C_u$ with $z \neq m$ CNs. Then, the CCS corresponding to $C_u$ is nonempty.*

*Proof.* Consider the set of VNs given by $\mathcal{N}_V(C_u)$. If every variable node in this set is only adjacent to CNs in $C_u$, then the Tanner graph is not connected, a contradiction to our assumption of connectedness. We conclude that the CCS corresponding to $C_u$ has at least one element. □

Recall that our experimental results will focus on $(j, k)$-regular and AB-LDPC codes. In the next section, we will be interested in optimizing clusters based on the number of edges incident to the corresponding CCSs. However, since $(j, k)$-regular and AB-LDPC codes have regular VN degree, examining the size of a CCS for these classes is equivalent to examining the number of edges adjacent to a CCS. We present below results on the size of a CCS for each class of LDPC codes.

**Theorem 1.** *Let $j, k \geq 2$, and $z \geq 1$ be integers. Suppose $G$ is a $(j, k)$-regular Tanner graph, and let $C_u$ be a cluster of CNs of size $|C_u| = z$ in $G$. If $|\mathcal{N}_V(C_u)| = v$, then the number of variable nodes in $W$, the CCS corresponding to $C_u$, is bounded as follows.*

$$v - \left\lfloor \frac{kz}{j} \right\rfloor \leq |W| \leq \min\{jv - kz, v\}.$$

*Proof.* It is straightforward to see that there are $jv - kz$ edges in $G$ that are incident to $\mathcal{N}_V(C_u)$ but not to $C_u$. In other words, there are $jv - kz$ edges exiting the subgraph induced by $C_u \cup \mathcal{N}_V(C_u)$.

Consider the CCS $W$, the subset of $\mathcal{N}_V(C_u)$ comprised of all variable nodes incident to an exiting edge. The minimum size of $W$ corresponds to the case in which the exiting edges are concentrated at a few variable nodes in $\mathcal{N}_V(C_u)$. That is, $|W| \geq \lceil (jv - kz)/j \rceil = \lceil v - (kz/j) \rceil = v - \lfloor kz/j \rfloor$. On the other hand, the maximum size of $W$ corresponds to the case where the exiting edges are spread across as many variable nodes as possible: $|W| \leq \min\{jv - kz, v\}$.

□

**Remark 1.** *For fixed $j, k$, and $z$, the upper and lower bounds given in Theorem 1 are increasing functions of $v$. In other words, the more neighbors a cluster has, the larger the maximum possible size of $W$ (and resulting $j|W|$ CCS edges), and the larger the minimum possible size of $W$ (and $j|W|$). It is important to note that we are not claiming that a higher number of neighbors necessitates a larger CCS. Rather, the shifting window of possible CCS sizes suggests a trend, even if a strict increase does not hold universally.*

Because they are $(3, p)$-regular, Theorem 1 gives bounds on the size of CCSs in $(3, p)$ AB-LDPC codes as well. However, given the added structure of an AB-LDPC code Tanner graph, we may conclude more, as shown in the following theorem, whose proof is provided in the supplementary material Section S.1.

**Theorem 2.** *Consider an AB-LDPC code defined by a parity-check matrix $\mathbf{H}(3, p)$, let $C_u$ be a cluster of size $1 \leq z \leq p$, and let $W$ be the CCS corresponding to $C_u$.*

*(i)* $|W| \geq ((1 + 2p)z - z^2)/4$.

*(ii) If all CNs in $C_u$ belong to the same row group of $\mathbf{H}(3, p)$, $|W| = zp$.*

*(iii) If $3 \leq z < p$ and every CN in $C_u$ belongs to at least one 6-cycle such that the other two CNs in the 6-cycle also belong to $C_u$, $|W| \leq zp - z$.*

The bounds in Theorem 2 should be compared with those of Theorem 1. Indeed, we find that Theorem 2(i) gives a tighter lower bound for smaller values of $\mathcal{N}_V(C_u)$, and Theorem 2(iii) is a tighter upper bound for a particular type of cluster and larger values of $\mathcal{N}_V(C_u)$. Theorem 2(ii), as an exact result, gives a clear improvement on Theorem 1 for a particular type of cluster.

The CNs of the cluster discussed in Theorem 2(ii) belong to the same row group in the corresponding parity-check matrix; we call CNs that are given by subsequent rows in the parity-check matrix *contiguous*. In comparison, not all CNs of the cluster discussed in Theorem 2(iii) are contiguous, since a 6-cycle must span three distinct row groups of a $\mathbf{H}(3, p)$ AB-LDPC code [27, 6]. By comparing the bounds on CCSs given in Theorem 2(ii) and (iii), we see that choosing non-contiguous CNs that comprise 6-cycles is guaranteed to lower the size of the corresponding CCS. Observing that the number of edges incident to a CCS $W$ in a $(3, p)$-regular graph is equal to $3|W|$, we conclude that in case of $(3, p)$ AB-LDPC codes, any cluster selection scheme should ensure that clustered CNs are not contiguous.

# 4   Cluster Optimization

In this section, we propose a cluster optimization strategy, with the goal of minimizing the number of edges connected to each CCS in a Tanner graph. Indeed, these edges are responsible for the propagation of messages between clusters, and consequently for the resulting dependencies between clusters. Let $E(C_u, W)$ be the set of edges that connect $C_u$ to its CCS $W$, $E(\bar{C}_u, W)$ the set of edges connecting $\bar{C}_u$ to $W$, and $\zeta(C_u) \triangleq |E(\bar{C}_u, W)| + |E(C_u, W)|$ the total number of edges by which $C_u$ is connected to $\bar{C}_u$ via $W$. Recall from Section 3 that in the case of $(j, k)$-regular Tanner graphs, minimizing $\zeta(C_u)$ is equivalent to minimizing the size of a CCS, since $\zeta(C_u) = j|W|$. We cluster the set of CNs of a Tanner graph via a greedy iterative process: the $z$ CNs of the first cluster, $C_1^*$, are chosen optimally from the set of all CNs. Each subsequent cluster is then chosen optimally from the set of remaining check nodes, with the last cluster consisting of the leftover CNs. Let $1, \ldots, u_{\lceil \frac{m}{z} \rceil}$ denote the indices of the clusters. Formally, this multi-step optimization is given by

$$C_e^* = \underset{C_e \subseteq C \setminus C_{e-1}^* \cup \cdots \cup C_1^*, \; |C_e| = z}{\arg\min} \zeta(C_e), \tag{5}$$

where $e \in 1, \ldots, \lceil \frac{m}{z} \rceil - 1$ denotes the index of the optimization step, $C_e$ represents a cluster with index $e$, and $C_e^*$ denotes the optimal cluster obtained in step $e$. The final cluster is obtained automatically, and hence excluded from (5).

Note that the complexity of the optimization in (5) grows exponentially with $m$, as we need to search over $\binom{m - z(e-1)}{z}$ possible cluster choices in each optimization step. To overcome this, we propose a more computationally feasible cluster optimization approach based on our observation that the lower bounds in Theorems 1 and 2 corresponds to a maximization of cluster-internal cycles: indeed, suppose that in each step, we cluster CNs so that the subgraph induced by the cluster and its neighbors contains as many cycles as possible. Such a cluster will have fewer neighbors compared to one that does not induce cycles. In turn, the maximum possible size of the corresponding CCS (and the number of exiting edges) will likely be reduced (see Remark 1). Thus, the cluster optimization approach based on cycle maximization is given by

$$\tilde{C}_e^* = \underset{C_e \subseteq C \setminus \tilde{C}_{e-1}^* \cup \ldots \cup \tilde{C}_1^*, \; |C_e| = z}{\arg\max} \eta_\kappa(C_e), \tag{6}$$

where $\tilde{C}_e^*$ denotes a cycle-maximized cluster, and $\eta_\kappa(C_e)$ denotes the number of length-$\kappa$ cycles in the graph induced by $\mathcal{N}_V(C_e) \cup C_e$. The possible choices of $\kappa$ depends on the girth of the family of codes considered; the smaller the choice of $\kappa$, the lower the optimization complexity, as larger cycles are more difficult to enumerate. In case of LDPC codes, optimized cycle detection algorithms based on message-passing have complexity of $\mathcal{O}(gE^2)$, where $g$ is Tanner graph's girth and $E$ is the total number of edges [17]. In a $(3, p)$ AB-LDPC code, whose graph has girth 6, we choose $\kappa = 6$. We present Algorithm 3 as a method for generating cycle-maximized clusters.

The algorithm inputs are a parity-check matrix $\mathbf{H}$, the cluster size $z$ and the length $\kappa$ of the cycles to be detected. Let $T$ be the total number of $\kappa$-cycles in $G_\mathbf{H}$. For each $x \in \{1, \ldots, T\}$, let $S_x = \{c_1, \ldots, c_{\kappa/2}\}$ denote the set of $\kappa/2$ CNs in the $\kappa$-cycle indexed by $x$, and define $\mathscr{C} = \{S_1, \ldots, S_T\}$.

---

**Algorithm 3:** Optimized clustering via cycle maximization

---

**Input** : $\mathbf{H}$, $z$, $\kappa$

**Output:** Set of cycle-maximized clusters $\tilde{C}_1^*, \ldots, \tilde{C}_{\lceil m/z \rceil}^*$

1   Initialize $\mathscr{C} \leftarrow \emptyset$, $x \leftarrow 1$

2   Run a suitable cycle detection algorithm on $G_{\mathbf{H}}$

3   **for** *each detected $\kappa$-cycle in $G_{\mathbf{H}}$* **do**

4      |   Determine $S_x$

5      |   Store $S_x$ in $\mathscr{C}$

6      |   $x \leftarrow x + 1$

7   **end**

8   **for** *$\kappa$-cycles $1, \ldots, T$* **do**

9      |   Determine the CN $c^* \in C \setminus \tilde{C}_{e-1}^* \cup \ldots \cup \tilde{C}_1^*$ appearing most frequently in $\mathscr{C}$

10     |   Find the $K$ sets $S_1, \ldots, S_K \in \mathscr{C}$ containing $c^*$

11     |   $\mathcal{C} \leftarrow S_1 \cup \cdots \cup S_K \setminus \tilde{C}_{e-1}^* \cup \ldots \tilde{C}_1^*$

12     |   **if** $|\mathcal{C}| \geq z$ **then**

13     |      |   Select $z$ CNs including $c^*$ from $\mathcal{C}$ and store them in $\tilde{C}_e^*$

14     |   **end**

15     |   **else**

16     |      |   Select $z - |\mathcal{C}|$ CNs randomly from $C \setminus \tilde{C}_{e-1}^* \cup \cdots \cup \tilde{C}_1^* \cup \mathcal{C}$ and store them in $\mathcal{C}'$

17     |      |   $\tilde{C}_e^* \leftarrow \mathcal{C} \cup \mathcal{C}'$

18     |   **end**

19   **end**

---

In Line 13, the algorithm selects $z$ distinct CNs, including $c^*$, appearing in $S_1, \ldots, S_K$, where $K$ is the total number of sets $S_x$ containing $c^*$. Note from Algorithm 3 that the optimization complexity depends on $T$, which is expected to be much smaller than $\binom{m-z(e-1)}{z}$ for sparse $\mathbf{H}$. For example, in a $(3, p)$ AB-LDPC code, there are only $T = p^2(p-1)$ 6-cycles in the Tanner graph [6]. In Section 5 we consider the case where $p \geq 5$ and $z = 7$. For these parameters and for $e = 1$, the number of cluster choices using the optimization in (5) is $\binom{\gamma p}{z} = \mathcal{O}(p^z)$ as opposed to $p^2(p-1)$ using (6).

# 5   Experimental Results

In this section, we perform learned sequential decoding by employing cluster-based Q-learning with contiguous, random, and (cycle-)optimized clusters.

The resulting schemes are denoted by MQC, MQR, and MQO, respectively. As a benchmark, we compare these three schemes to that wherein the MAB problem is solved by letting the action-value function be equal to the Gittins index (GI): roughly speaking, the GI represents the long-term expected reward for taking a particular action in a particular state under the assumption that the arms are independent bandit processes [25, 11]. We denote this benchmark scheme by MGI. We then utilize each of these schemes for sequential decoding of both $(3, 7)$ AB-LDPC and random $(3, 6)$-regular LDPC codes. Specifically, Algorithm 2 is used to implement the three cluster-based learning approaches mentioned above.

As a first step towards other RL approaches, we have implemented a NS decoder based on Thompson Sampling [22], NS-TS, which performs better than flooding, but worse than our proposed Q-learning scheme. Our NS-TS scheme tracks the densities of the messages via a Gaussian approximation and uses the MSE $(m'_{a \to v} - m_{a \to v})^2$ as a non-central chi-square distributed reward, sampled in each sequential decoding step.

For both considered codes, we let the learning parameters be as follows: $\alpha = 0.1$, $\beta = 0.9$, $\epsilon = 0.6$, $z = 7$, $M = 4$, $\ell_{\max} = 25$, and $|\mathscr{L}| = 1.25 \times 10^6$. We note that this choice of $|\mathscr{L}|$ is sufficiently large in order to accurately estimate the action-value function. For clustered Q-learning, training is accomplished by appropriately updating the CN scheduling step of Algorithm 2 for each code, and the corresponding action-value functions are used to perform MQC, MQR, and MQO. We compare the performances of these MAB-NS schemes with the existing non-Q-learning-based BP decoding schemes of flooding, MGI, and NS for $\ell_{\max} = 25$.

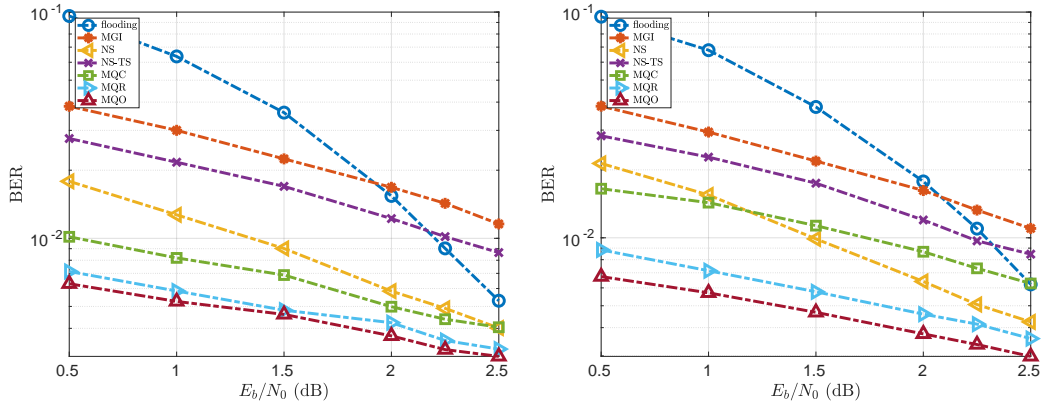

**Figure 2:** BER results using different BP decoding schemes for a $(3,6)$-regular LDPC (left figure) and $(3,7)$ AB-LDPC code (right figure, code is lifted), with block length $n = 196$.

The bit error rate (BER), given by $\Pr[\hat{x}_i \neq x_i]$, $i \in [[n-1]]$, vs. channel signal-to-noise ratio (SNR) $E_b/N_0$ in dB of $(3,6)$-regular and $(3,7)$ AB-LDPC codes using these decoding techniques are shown in Fig. 2 on the left- and right-hand side, respectively.

The experimental results reveal that MQO is superior to the other decoding schemes in terms of BER performance for both codes. In Table 1, we compare the average number of CN to VN messages propagated in the considered decoding schemes to attain the results in Fig. 2. The numbers without (resp. with) parentheses correspond to the $(3,6)$-regular (resp. $(3,7)$ AB-) LDPC code. We note that the MQO scheme, on average, generates a lower number of CN to VN messages when compared to the other decoding schemes.

Moreover, in contrast to NS, RL-based schemes avoid the computation of residuals in real-time, providing a significant reduction in message-passing complexity for short LDPC codes.

**Table 1:** Average number of CN to VN messages propagated in various decoding schemes for a $(3,6)$-regular $((3,7)$ AB-) LDPC code to attain the results shown in Fig. 2.

| SNR (dB) | 0.5 | 1 | 1.5 | 2 | 2.5 |
|---|---|---|---|---|---|
| flooding | 27040 (27913) | 21716 (24532) | 13327 (16098) | 6316 (8117) | 2500 (3064) |
| MGI | 308 (365) | 275 (324) | 234 (281) | 210 (247) | 174 (205) |
| NS | 293 (362) | 266 (332) | 229 (279) | 203 (245) | 173 (203) |
| MQC | 351 (411) | 299 (369) | 266 (313) | 218 (264) | 190 (215) |
| MQR | 310 (368) | 275 (326) | 235 (286) | 214 (244) | 182 (214) |
| MQO | 264 (367) | 237 (333) | 209 (283) | 181 (243) | 163 (208) |

# 6 Conclusions

We presented a novel RL-based decoding framework to optimize the scheduling of BP decoders for short-to-moderate length LDPC codes. The main ingredient is a new state space clustering approach to significantly reduce the learning complexity. Experimental results show that optimizing the cluster structure by maximizing cluster-internal short cycles provides marked performance gains when compared with both previous scheduling schemes and clustered Q-learning with non-optimized clusters. These gains include lowering both BER and message-passing complexity. As Bayesian inference over graphical models is at the core of many machine learning applications, on-going work includes extending our RL approach to other applications involving BP-based message passing over a factor graph defined by an underlying probabilistic model.

# Broader Impact

The proposed approach outlines a comprehensive RL framework for low-complexity, low-latency, decoding of short-to-moderate length LDPC codes. This approach has broader technological and societal impacts through several complementary mechanisms, elaborated on as follows. One broader impact is on the advancement of information technology – and its benefits to society – through newly established theory and practice in the area of RL-based error correction for linear graph-based codes. As global per-capita information usage continues to grow significantly over the next decade, considerable improvements in error correcting codes will be required in order to satisfy the high-throughput, low-latency requirements of emerging communication systems, while ensuring scalable service requirements for a host of heterogeneous devices. Our application of RL provides a significant gain in performance while reducing decoding complexity for linear graph-based codes. As a consequence, the results of this work have the potential to materially impact our daily lives. In addition, the results presented in this work have the potential to have a significant transformative impact on other critical applications employing low-latency, reliable networked communication. Among many others, these include applications in the fields of healthcare, environmental monitoring, and finance.

# Acknowledgment

Funding in direct support of this work: NSF grant ECCS-1711056 and by the Combat Capabilities Development Command Army Research Laboratory under Cooperative Agreement Number W911NF-17-2-0183. We thank the anonymous reviewers and the meta reviewer for their comments and suggestions to improve the paper.

## Footnotes

[1]Note that, in recent literature such as [16, 30], MAB refers to a problem where actions do not change the state of the environment. However, in this paper, we refer to MAB as the traditional bandit problem discussed in [7, 11] where an agent's action does indeed change the state of the environment.

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
