[Supplementary Material]

# Supplementary Material for: Learning to Decode: Reinforcement Learning for Decoding of Sparse Graph-Based Channel Codes

## S.1 Proof of Theorem 1

The proofs are ordered according to the claims in the theorem as follows.

1. There are $p$ VN neighbors of a CN $c \in C_u$, and none of these VN neighbors can share another CN neighbor due to the absence of 4-cycles. Thus, the number of VNs in $\mathcal{N}_V(c)$ that have no neighbors outside of $C_u$ is at most $(z-1)/2$ (each of these VNs has two other neighbors in the remaining $z-1$ CNs of $C_u$, and all are distinct). Thus, there are at least $p - (z-1)/2$ VNs adjacent to $c$ that have at least one neighbor outside of $C_u$. This is true for all choices of $c$. Since we may be counting each of these VNs up to two times, there are at least $(p - [(z-1)/2])z/2$ VNs in $W$. The result follows.

2. If all the rows corresponding to the CNs of cluster $C_u$ are in the same row group, then no two CNs in $C_u$ will have any VNs in common. Hence, each VN in $\mathcal{N}_V(C_u)$ must also be adjacent to $\bar{C}_u$, implying that $\mathcal{N}_V(C_u)$ is a CCS with $|\mathcal{N}_V(C_u)| = |C_u|p = zp$.

3. Suppose that the CNs in $C_u$ form triples, and let $\phi = \{\{c_1, c_2, c_3\}, \ldots, \{c_{z-2}, c_{z-1}, c_z\}\}$, $|\phi| = z/3$, be a set of all these triples. Suppose that each CN triple in $\phi$ is associated to a non-overlapping 6-cycle. Since, in this case, there will be $|\phi|$ non-overlapping 6-cycles in the Tanner graph induced by $C_u \cup \mathcal{N}_V(C_u)$, and each 6-cycle has 3 VNs, the number of VNs that have degree 2 with respect to $C_u$ will be $|\phi| \times 3 = z$. Note that there are $zp$ edges emanating from $C_u$ since each CN degree is $p$. Hence, in the worst case, there will be $zp - z$ distinct VNs in $\mathcal{N}_V(C_u)$, which is also a CCS since each of these degree 3 VNs are also connected to $\bar{C}_u$. $\square$

# S.2 Algorithm 1

---

**Algorithm 1:** MAB-NS for LDPC codes

---

**Input** : **L**, **H**
**Output:** reconstructed signal

1 Initialization:
2    $\ell \leftarrow 0$
3    $m_{c \rightarrow v} \leftarrow 0$                          `// for all CN to VN messages`
4    $m_{v \rightarrow c} \leftarrow L_v$                         `// for all VN to CN messages`
5    $\hat{\mathbf{L}}_\ell \leftarrow \mathbf{L}$
6    $\hat{\mathbf{S}}_\ell \leftarrow \mathbf{H}\hat{\mathbf{L}}_\ell$
7 **foreach** $a \in [[m]]$ **do**
8    |  $s_\ell^{(a)} \leftarrow g_M(\hat{s}_\ell^{(a)})$                   `// `$M$`-level quantization`
9 **end**
   `// decoding starts`
10 **if** *stopping condition not satisfied or* $\ell < \ell_{\max}$ **then**
11    |  $s \leftarrow$ index of $\mathbf{S}_\ell$
12    |  select CN $a$ according to an optimum scheduling policy
13    |  **foreach** *VN* $v \in \mathcal{N}(a)$ **do**
14    |    |  compute and propagate $m_{a \rightarrow v}$
15    |    |  **foreach** *CN* $c \in \mathcal{N}(v) \setminus a$ **do**
16    |    |    |  compute and propagate $m_{v \rightarrow c}$
17    |    |  **end**
18    |    |  $\hat{L}_\ell^{(v)} \leftarrow \sum_{c \in \mathcal{N}(v)} m_{c \rightarrow v} + L_v$ `// update LLR of `$v$
19    |  **end**
20    |  **foreach** *CN* $j$ *that is a neighbor of* $v \in \mathcal{N}(a)$ **do**
21    |    |  $\hat{s}_\ell^{(j)} \leftarrow \sum_{v' \in \mathcal{N}(j)} \hat{L}_\ell^{(v')}$
22    |    |  $s_\ell^{(j)} \leftarrow g_M(\hat{s}_\ell^{(j)})$             `// update syndrome `$\mathbf{S}_\ell$
23    |  **end**
24    |  $\ell \leftarrow \ell + 1$                       `// update iteration`
25 **end**

---

# S.3 Details of Algorithm 2

---

**Algorithm 2:** Clustered Q-learning

---

**Input** : $\mathscr{L}$, $\mathbf{H}$
**Output:** Estimated $Q_{\ell_{\max}}(s_u, a_u)$ for all $u$

1   Initialization: $Q_0(s_u, a_u) \leftarrow 0$ for all $s_u$, $a_u$ and $u$
2   **for** *each* $\mathbf{L} \in \mathscr{L}$ **do**
3     $\ell \leftarrow 0$
4     $\hat{\mathbf{L}}_\ell \leftarrow \mathbf{L}$
5     $\hat{\mathbf{S}}_\ell \leftarrow \mathbf{H}\hat{\mathbf{L}}_\ell$
6     **foreach** $a \in [[m]]$ **do**
7       $s_\ell^{(a)} \leftarrow g_M(\hat{s}_\ell^{(a)})$              // $M$-level quantization
8     **end**
9     **while** $\ell < \ell_{\max}$ **do**
10       schedule CN $a_u$ according to an $\epsilon$-greedy approach
11       select $u$ as cluster index of CN $a_u$
12       $\mathbf{S}_\ell^{(u,z)} \leftarrow s_\ell^{(j_1)}, s_\ell^{(j_2)}, \ldots, s_\ell^{(j_z)}$ // CN indices $j_1, j_2, \ldots, j_z \in \{0, \ldots, m-1\}$ are in ascending order for MQC, and randomly ordered for MQR. For MQO, their ordering depends on the underlying cluster $\tilde{C}_u^*$
13       $s_u \leftarrow$ index of $\mathbf{S}_\ell^{(u,z)}$
14       **foreach** *VN* $v \in \mathcal{N}(a_u)$ **do**
15         compute and propagate $m_{a_u \to v}$
16         **foreach** *CN* $c \in \mathcal{N}(v) \setminus a_u$ **do**
17           compute and propagate $m_{v \to c}$
18         **end**
19         $\hat{L}_\ell^{(v)} \leftarrow \sum_{c \in \mathcal{N}(v)} m_{c \to v} + L_v$ // update LLR of $v$
20       **end**
21       **foreach** *CN* $j$ *that is a neighbor of* $v \in \mathcal{N}(a_u)$ **do**
22         $\hat{s}_\ell^{(j)} \leftarrow \sum_{v' \in \mathcal{N}(j)} \hat{L}_\ell^{(v')}$
23         $s_\ell^{(j)} \leftarrow g_M(\hat{s}_\ell^{(j)})$            // update syndrome $\mathbf{S}_\ell$
24       **end**
25       $s_u' \leftarrow$ index of updated $\mathbf{S}_\ell^{(u,z)}$
26       $R_\ell(s_u, a_u, s_u') \leftarrow$ highest residual of CN $a_u$
27       compute $Q_{\ell+1}(s_u, a_u)$
28       $\ell \leftarrow \ell + 1$                  // update iteration
29     **end**
30   **end**

---