[Reviews · NeurIPS 2020]

Review 1

Summary and Contributions: This paper focuses on bandit approaches to decoding LDPC codes, building up the emerging area of using machine learning techniques in communications, where training data is "infinite" and performance metrics are very clear. The idea is to learn a good sequential scheduling policy for the order in which different check nodes are operated on (and presumably not to get stuck in combinatorial structures such as stopping sets), to improve decoding performance. This is in contrast to other work in this broad area that focuses on finding coding schemes, rather than improving decoding algorithms. Comparisons to other decoders shows significant improvements. It seems the authors have taken all the reviewer comments to heart and will have an improved paper.

Strengths: * ML for communications is a nice area of exploration and such investigations push the scope of neurIPS to settings where performance criteria are very clear * Performance improvement over BP is compelling and practically could be important * Broader Impacts section well written and broader impacts themselves are quite positive

Weaknesses: * Although the general ML for communications area is relevant for neurips, this particular paper seems too specific to be of general interest, and may fit better in a venue like IEEE Trans. Commun. * There is some confusion between multi-armed bandit and reinforcement learning in this work * No explanation/intuition of why this dynamic scheduling helps is given: some greater insight from [3] would have set the stage better for this work

Correctness: I went over the paper fairly closely and it looks generally correct. Empirical evaluation is quite standard in the communications field.

Clarity: As noted above, some initial intuition as to why scheduling helps (from [3]) would help the reader understand all that is to follow.

Relation to Prior Work: A greater discussion of how this work differs from other ML-for-comms work would help the general reader understand that this paper is focused on the internal operation of the decoding algorithm rather than other work on designing feedback coding schemes (deepcode), etc.

Reproducibility: Yes

Additional Feedback:


Review 2

Summary and Contributions: The paper proposes a novel decoding method for LDPC codes based on reinforcement learning. The proposed method determines check node scheduling based on Q-learning. Some numerical experiments shown in Section 5 indicates that the proposed method provides faster convergence compared with the conventional ones.

Strengths: As far as I know (and as the authors claim), this paper is the first work to employ reinforcement learning to improve decoding performance of BP decoding. I thus think that the paper has enough novelty. The main idea appears quite natural and reasonable to me. The numerical results shown in Section 5 supports the effectiveness of the idea.

Weaknesses: The reward function (2) without detailed explanation (I guess "informative messages" are more rewarded). This part (i.e., justification of the reward function) may be important to understand the proposed scheme. In Fig 2, I would like to see high SNR (waterfall) performance as well. Compared with the naive BP (BP), the decoder using the proposed scheme requires to memorize Q-table to select a CN. And, time complexity of selecting a CN (line 12, Algorithm 1) can be explicitly described in the text. Such overheads (compared with a naive flooding BP) should be written for fair comparison. ========== After Authors' Rebuttal ========== I read the authors' response and fully understand their explanation.

Correctness: The empirical methodology used in the paper seems correct to me.

Clarity: I think the paper is well written.

Relation to Prior Work: I think the description in the paper is enough.

Reproducibility: Yes

Additional Feedback: It would be nice to see "interpretation of the learned schedules". Some good CN selection rules have been obtained in the experiments. If the learned schedule follows a certain simple rule, I would like to know it, and such an observation may lead to another optimal scheduling without a learning process.


Review 3

Summary and Contributions: This paper aims at improving LDPC node-wise scheduling decoding performance, by forming the node selection problem as a multi-arm bandit (MAB) problem, which can be solved by Q-learning. This paper also uses clustered Q-learning and cluster connecting sets, to reduce the unavoidable complexity coming from long block length nature of channel coding. The proposed method shows better BER performance and less complexity comparing to a few solid existing benchmarks.

Strengths: LDPC code is an indispensable building block for LTE/5G communication systems, a more efficient and accurate decoding algorithm is impactful for current communication systems. Node-wise scheduling (NS) is known to improve decoding efficiency, yet incurs more complexity. Using Q-learning Table the computation complexity improves, which makes the NS-based method become viable. The long block length nature of LDPC code, makes the number of state exponential. The author uses clustered based method to reduce the number of potential state. Using the structure of LDPC code, the cluster-connected set (CCS) is bounded, and can be further optimized via cycle maximization. The proposed method shows better BER performance and less complexity, on not-very-long LDPC code.

Weaknesses: However, the long block length nature of LDPC code incurs a very special clustering and optimization method to migigate the large state space problem. The proposed method is very specific to channel coding, which doesn't have enough interest to the general audience. Moreover, for very large block length (e.g. LDPC code can be longer than 1000, due to block length gain of channel coding), the proposed algorithm could still suffer from large state space. Some further complexity reduction methods might be needed. Furthermore, the Q-learning with quantization is a good first try, yet there exists quite a few RL algorithm which can better support this MAB for LDPC NS decoding problem.

Correctness: Yes.

Clarity: Yes

Relation to Prior Work: Yes.

Reproducibility: No

Additional Feedback:


Review 4

Summary and Contributions: The paper introduce a method for decoding short block codes. The method utilize a sequential update policy for the check nodes by model it as bandits problem. Furthermore, they use clustering approach to reduce the learning complexity.

Strengths: The ideas in the paper to learn the scheduling of the check node is very interesting. Moreover, the learning of the scheduling updates is novel.

Weaknesses: I have a major concern about the results: 1. Please use the standard metrics to evaluate the model, i.e. please use BER/SNR figure. It is very hard to judge what is the improvement with the proposed method with BEP/SNR. 2. Please provide more results for other short block codes (for example - BCH), and compare it to other SOTA results as in previous papers: "I. Be’ery, N. Raviv, T. Raviv, and Y. Be’ery. Active deep decoding of linear codes. IEEE Transactions on Communications" "Nachmani, Eliya, and Lior Wolf. "Hyper-graph-network decoders for block codes." Advances in Neural Information Processing Systems. 2019."

Correctness: seems correct

Clarity: yes

Relation to Prior Work: Yes, the paper show what is the difference between the proposed method to other learnable decoders.

Reproducibility: Yes

Additional Feedback:

[Author Response · NeurIPS 2020]

We thank the reviewers for their thoughtful comments and suggestions. Below, we address the reviewers' main concerns and recommendations; our responses will be incorporated in the final version of the paper.

**Relevance of the work to NeurIPS [R1, R3]:** Our work is focused on a channel coding setting, and the realization of our cluster optimization scheme is geared specifically toward decoder scheduling of sparse graph-based codes. Note that channel coding applications have been of interest to the NeurIPS community previously (e.g. [1, 2, 3, 4] recently). On a high level, as Bayesian inference over graphical models is at the core of many machine learning applications, we believe that learned scheduling of belief propagation (BP) may be similarly applied to BP-based message passing over a factor graph defined by an underlying probabilistic model, making our approach very relevant to NeurIPS.

**Providing more intuition [R1, R2]:** We note here that our algorithm is dynamic, and depends both on the graph structure and on received channel values: thus, the schedule realization may change for subsequent transmissions, and will outperform a schedule that is fixed in advance. NS relies on the intuition that in loopy BP, the higher the residual of a CN, the further that portion of the graph is from convergence. Hence, scheduling CNs with higher residuals is expected to lead to faster decoder convergence. As NS follows a fixed greedy schedule, there exists a non-zero probability that initially correct, but unreliable, bits are wrongly corrected into an error that is propagated in subsequent iterations. In contrast, our proposed scheme based on Q-learning allows some room for exploration (not just exploitation, as in NS) by scheduling the CN with the highest expected *long-term* residual, mitigating such a potential error propagation.

**Updates to simulation results [R2, R4]:** We have added results for higher SNR (see Fig. (a)) for $(3, 6)$-regular LDPC codes, which indicate that the MQO scheme significantly outperforms the non-RL decoding schemes. The results for $(3, 7)$ AB codes are given in Fig. (b), showing similar results. We also simulated a $(63, 51)$ BCH code. However, since this is a high density parity check code, the training complexity is higher than for LDPC codes. As a snapshot, we obtained a result for an SNR of 3.5 dB with a BER of $10^{-2}$ for MQO and of $1.3 \cdot 10^{-2}$ for the hypernetwork decoder of [4], showing again the gain achieved by our RL approach. We have also adjusted the metrics of the paper from BEP to BER, as suggested by R4.

**Complexity comments [R2, R3]:** We note that the time complexity for selecting a CN in line 12 of Alg.1 grows linearly with the total number of CNs, as opposed to being zero for BP flooding. This overhead will be discussed more explicitly in the final paper. The concerns of R3 regarding the comparatively larger state space of longer block lengths, even with clustering, is an area of ongoing work. However, we believe that the substantial gains of our optimized clustering method demonstrated at lower block lengths are promising, and remain an important contribution in and of themselves. As a first approach to mitigating complexity at longer block lengths, we have implemented a Thompson sampling (TS) approach (see Fig. (a)). Another approach is to approximate the Q-table via a neural network.

**Distinction from previous work [R1]:** We thank R1 for the valuable suggestion. Our work also differs from the vast majority of works (including those cited below) in that our decoder is not based on deep learning.

**Alternative RL algorithms [R3]:** As a first step towards other RL approaches, we have implemented a decoder based on TS (see Fig. (a)), which performs better than flooding, but worse than our proposed Q-learning scheme. In this TS-based scheme we track the densities of the messages via a Gaussian approximation and use the MSE $(m'_{a \to v} - m_{a \to v})^2$ as a non-central chi-square distributed reward, sampled in each sequential decoding step.

### References

[1] Rishidev Chaudhuri and Ila Fiete. Bipartite expander Hopfield networks as self-decoding high-capacity error correcting codes. In *Advances in Neural Information Processing Systems*, pages 7688–7699, 2019.

[2] Yihan Jiang, Hyeji Kim, Himanshu Asnani, Sreeram Kannan, Sewoong Oh, and Pramod Viswanath. Turbo autoencoder: Deep learning based channel codes for point-to-point communication channels. In *Advances in Neural Information Processing Systems*, pages 2758–2768, 2019.

[3] Hyeji Kim, Yihan Jiang, Sreeram Kannan, Sewoong Oh, and Pramod Viswanath. Deepcode: Feedback codes via deep learning. In *Advances in Neural Information Processing Systems*, pages 9436–9446, 2018.

[4] Eliya Nachmani and Lior Wolf. Hyper-graph-network decoders for block codes. In *Advances in Neural Information Processing Systems*, pages 2329–2339, 2019.


[Meta-Review · NeurIPS 2020]

This paper proposes application of reinforcement learning, in particular Q-learning, to determine the check-node (CN) scheduling policy in BP decoding of short LDPC codes. It is in contrast to other works in the broad area of machine learning applications to coding which focus on finding coding schemes or "deep unfolding" of iterative decoders. Discretization of state space and clustering of CNs are introduced to avoid explosion of the state space size and learning complexity. The experiments in Section 5 show improvement by the proposal both in terms of error probability and the number of CN-to-variable-node message propagation. The reviewers rated this paper favorably, especially with emphasis on the novelty. They are also satisfied with the author response. I would thus like to recommend acceptance of this paper. I would also appreciate it if the authors take into account the following items, as well as the review comments. - Use of the term "bandit": As Reviewer #1 mentioned, the reader may think that there is some confusion in this paper between bandit and reinforcement learning. Although it seems that the authors use the term "bandit" in accordance with reference [6], where it is used to refer to a problem where the environment consists of N statistically independent processes of state transition and reward generation, each with its own state space, the use of the term "bandit", regarded standard nowadays, refers to a problem where the environment has only a single state so that the action set and the reward processes in the future are not affected by the decisions today. See, e.g., Section 1.1.2 of Lattimore and Szepesvari, Bandit Algorithms, Cambridge University Press, 2020 (accessible via https://tor-lattimore.com/downloads/book/book.pdf). A clarification regarding this point should be useful. - Line 132: The use of the term "value" here is somehow confusing, as it refers here to the numerical value representing the aggregated log likelihood ratios, whereas the term would generally refer in the standard reinforcement learning framework to (estimated) expected return of a state or a state-action pair. - Line 305: The quantity Pr[\hat{x}\not=x] corresponds not to the bit error rate claimed here by the authors, but to the block error rate, as the boldface x and hat(boldface x) are defined to be the transmitted codeword and the reconstructed codeword in line 128 and in lines 130-131 of the paper, respectively. In order to avoid possible confusion between bit error rate and block error rate, a convention in communication engineering is to abbreviate them as BER and BLER, respectively, which the authors would like to adopt with an explicit definition. I also think it a good idea to mention that the target block error rate of 10e-1 has been employed in several radio network specifications, so that the shown range of error rates in Figure 2 is relevant.